# Barriers and Challenges for Visually Impaired Students in PE—An Interview Study with Students in Austria, Germany, and the USA

**DOI:** 10.3390/ijerph20227081

**Published:** 2023-11-19

**Authors:** Sebastian Ruin, Justin A. Haegele, Martin Giese, Jana Baumgärtner

**Affiliations:** 1Institute of Human Movement Science, Sport and Health, University of Graz, 8010 Graz, Austria; jana.baumgaertner@uni-graz.at; 2Center of Movement, Health, & Disability, Old Dominion University, Norfolk, VA 23529, USA; jhaegele@odu.edu; 3Department of Natural and Human Sciences, Heidelberg University of Education, 69120 Heidelberg, Germany; martin.giese@ph-heidelberg.de

**Keywords:** children with visual impairments, physical education, qualitative study, international, barriers, challenges, ideas of normality, typology

## Abstract

Physical education (PE) is an important part of school education worldwide, and at the same time, almost the only subject that explicitly deals with body and movement. PE is therefore of elementary importance in the upbringing of young people. This also applies to children with visual impairments. However, existing findings on participation and belonging in PE as well as on physical and motor development reveal that this group of children and adolescents is noticeably disadvantaged in this respect. Against this background, this paper aims to explore fundamental barriers and challenges across different types of schools, types of schooling, and countries from the perspective of visually impaired children. The qualitative interview study with 22 children with visual impairments at different types of schools in three countries (Austria, Germany, USA) reveals that none of the respondents could escape the power of social distinctions and related problematic and existing hierarchies. Hence, ideas of normality and associated values remain the main challenge for all of them. However, the type-forming analysis provides important insight across settings on how visually impaired children differ on this, allowing for greater sensitivity to the concerns of children with visual impairments.

## 1. Introduction

As a result of a global inclusion agenda and the worldwide acceptance of the UN Convention on the Rights of Persons with Disabilities (CRPD), education systems in States Parties are required to help overcome the exclusion of marginalized and discriminated groups of society. According to Article 24 (2a) of the CRPD “children with disabilities are not [to be] excluded from free and compulsory primary education, or from secondary education, on the basis of disability”. In international educational contexts, this statement is usually interpreted in such a way that the realization of the right to inclusion is tied to the fact that children with and without disabilities are taught jointly in the same class or environment [1]. The question of a mindful setting is considered a question of placement. The normative orientation of this educational policy is largely undisputed in the international discourse on inclusion in physical education (PE). As a result, segregating educational institutions worldwide tend to be actively disbanded to involve all pupils in general education, including physical education [2].

However, the degree to which integrated education is embraced, and segregated education disbanded, appears to be dependent on cultural context and impairment. For example, with regard to children with visual impairments (abbreviated below as CWVI), McLinden et al. [3] (p 180) state that for the Anglo-American context that “the majority of children and young people with vision impairments but no additional disabilities are now educated in mainstream settings.” This does not apply to Germany and Austria, where the interpretation of Article 24 (2a) of the CRPD is highly controversial and there is a vehement dispute over how the CRPD should be properly interpreted [1]. There are strong trends in maintaining special schools in both countries, especially with regard to sensory impairments.

Our goal is to contribute to the discussion of superficial issues of placement in order to transfer the argument to a structural level. Regardless of the setting (placement) in which CWVI are schooled, these children should be supported in such a way that they feel valued and involved in their respective PE lessons. (Inter)national studies, however, show that inclusive experiences are often hardly accessible to disabled students in PE [4]. Regarding research methodology, it should be noted that this typically involves research *about* students with disabilities [5]. Hence, research activities on PE refer to the perspectives of parents [6], nondisabled peers [7], or nondisabled teachers [4]. Such research approaches tend to emphasize the perspectives of nondisabled peers, parents, teachers, and experts, while the voices of disabled students are systematically ignored. This is problematic in that it limits our understanding of the thoughts, feelings, and experiences of these students, which are central to the design of mindful settings for disabled students [8]. Against this background, the following explanations are based on a subjective understanding of inclusion (feel included). Inclusion is understood as a subjective experience that refers to feelings of acceptance, value, and belonging [9]. Inclusion is not reduced to measuring whether a person is present in certain rooms (or school forms), but whether subjective feelings of acceptance, value, and belonging on an individual level actually arise. In the canon of research on disabled persons in PE, this approach mainly includes research on visually impaired students showing that many CWVI have extensive negative experiences in PE. For example, it is reported that PE is often unilaterally dominated by ableist body and performance standards that visually impaired students are often unable to meet in their self-awareness [9].

At the same time, peers and teachers appear to be indifferent to the needs of CWVI and show little willingness to reflect and adapt the teaching or social setting. Previous research indicates that this unwillingness seems to be linked by socially conveyed norms of what is perceived as normal and ableist ideas [10]. The micro level of interpersonal action is strongly formatted by normality discourses on the macro level, and this also infects teaching methodology [11]. This requires the perception of being different in CWVI and the assumption that they are located at (or even under) the bottom of the social hierarchy in the class [9]. Similar results are also evident in bullying research. That is, according to Ball et al. [12], CWVI often have bullying experiences and are excluded and isolated from activities by teachers. As we can see, there is a strong body of literature that shows that the implementation of mindful PE from the perspective of visually impaired students is linked to a variety of barriers on a societal level as well as on an interpersonal and emotional micro level and on the level of didactic decisions.

Against this background, the subjective perceptions of barriers and challenges of CWVI in PE are the focus of research. Identifying and deconstructing barriers and challenges is an important goal, as regular participation in physical activities can make a positive contribution to social, mental, and physical well-being [13]. Hence, this article aims to reconstruct subjective perceptions of barriers and challenges of visually impaired students in PE with regard to all three levels mentioned above (societal, interpersonal-emotional, and didactic) within the meaning of Article 8 of the CRPD—the article on awareness raising (awareness-raising). So far, however, we only have national research results with data collected in different types of schools, which have been considered independently of each other. To bridge this gap, in this article, we look at interviews from three different countries in three different types of schools. We use this perspective to expand our transnational understanding of the subjective perceptions of barriers and challenges of CWVI in PE. This might give first insights into assessing the importance of the respective school settings in different school systems in different countries. Ultimately, however, our aim is to identify exclusion processes on a structural level that underlie the surface phenomena.

## 2. Materials and Methods

To reconstruct the students’ perspectives on PE and associated barriers and challenges in terms of survey as well as analysis a qualitative approach was applied that seeks to access subjective viewpoints [14]. This also appears to be helpful in order to be sensitive to the specifics of the deliberately chosen different settings. The general aim was to reconstruct specific manifestations of social reality (the perspectives of CWVI, respecting their individual situation) in certain situations (the PE classes, they attend) [15]. The observation of these situational manifestations of social reality in deliberately contrastive settings was therefore not intended to reveal reality per se via a representative sample. Rather, the aim of the study was to identify structural similarities between the situational manifestations and to understand individual characteristics with regard to these structural similarities.

### 2.1. Sample

To address the question of whether certain challenges tend to be consistent across settings, or whether other challenges are more sporadic, the sample was deliberately chosen so that respondents came from three different countries, each with different school traditions, and each from different school settings. Therefore, the sample consists of respondents in Austria with a traditional, relatively conservative school system, in Germany with a still highly specialized and segregated school system for children with special needs, and in the USA with decades of experience in educating children with special needs in the general school system. In addition, three different types of schools were selected to cover a relatively broad spectrum in the sample as well: a vocational school with a special focus on children with special needs in Austria (7 respondents), a special high school for CWVI in Germany (8 respondents), and general high schools in the USA (7 respondents). The ages of the 22 total respondents ranged from 12 to 21 (with two respondents not providing age information), 10 of whom identified themselves as female, and 12 as male (see Table 1).

### 2.2. Data Collection and Analysis

After teachers and parents gave informed consent, semi-structured guided interviews were conducted with the respondents in the three settings. The interview guide included a general part about PE, its understanding, and the subjective description of the subject. This was followed by questions about the relationship with teachers and classmates and the emotional evaluation of these. Subsequently, questions were asked about performance expectations and the didactic design of the lessons. Finally, it was asked whether there was any experience with bullying. The first interviews were conducted in Germany in 2019 at the beginning of the COVID-19 pandemic face-to-face as interviews in pairs (with two exceptions caused by illness of the tandem twin) when the schools there were still open. With the restrictions and changes during the pandemic, subsequent interviews were conducted as single interviews by telephone in Austria in 2021 and by videoconference in the United States in 2022. In Austria and Germany, the interviews were conducted in German and after transcription translated into English by a professional language service for the analysis.

The (translated) interview transcripts were the basis for a qualitative text analysis [16]. The categories were elaborated in recognition of our theoretical and empirical prior knowledge and assumptions [17] outlined in the introduction which led to three main categories with subcategories elaborated in a deductively-inductively-process in the confrontation with the data (see Figure 1). Thereby, the subcategories are understood as evaluative categories (e.g., regarding the category “Feelings of (not) being normal” a respondent could be evaluated in three ways: the respondent (1) feels to be normal, (2) is ambivalent or (3) does not feel they are normal). This way we rated each respondent on each subcategory. In some cases, this was not possible due to a lack of information in this regard; in that case, we made a short note.

Based on the ratings for each respondent our aim was to identify a bigger transnational picture regarding differences and continuities between the respondents and the educational settings and contexts. Also, we sought to identify connections between the categories in individual contextualizations (providing short portraits of each respondent) as well as communalities within the whole sample. In doing so, we further searched for a possible typology of individual positionings within the complex relations investigated.

## 3. Results

The analysis led to differentiated perspectives on the main categories that differentiate the different levels of problematized aspects. These are found on societal (macro), interpersonal, and emotional (micro) as well as didactical (instructional) levels and are further differentiated in the assigned subcategories (see Table 2). In the following, these categories and subcategories are described based on findings across the sample, to provide a better understanding of the semantic content and possible expressions (3.1). Following this, the results of the type-forming analysis are presented, and the identified types are characterized (3.2).

### 3.1. Findings across the Sample by Categories

To provide a structured overview, in this chapter, the findings regarding the categories are described and detailed in terms of their semantic content. The description is sorted by the main and sub-categories (Figure 1).

#### 3.1.1. Societal Aspects—Discursive Ideas of Normal

Although not explicitly asked in the interviews, almost every interviewee talks about their feeling of being normal or not normal. Some interviewees relate this explicitly to certain ideas of how a normal person is characterized—other interviews show such a connection in a more implicit way. The more or less ever-present idea of normality tends to be oriented towards ableist concepts of the body and sports-related expectations, which are expressed via attributions such as “sporty” and “active” kids (Giorgio, GER, 00:08:25-1). Students’ emotions toward these ideas range from positive feelings about experiencing themselves as normal (“There were three exercises and I got the highest score there, so that was really cool”; Simon, AT, 00:17:17-9), to negative feelings associated with wishing one could be “a normal child” (Clara, AT, 00:11:18-9). While perceptions of being normal diverge across the sample, it seems almost common sense that deviation from the norm is perceived as more or less problematic. However, the identified problem varies here from a perceived “disadvantage” due to visual impairment (Giorgio; GER, 00:11:09-4) and an attributed inability to participate in certain sports, to a general societal lack of understanding of blind people (Lily, US, 00:33:50-2), to referring to other children with severe or multiple impairments as victims of bullying due to their great distance from what is considered normal (Timo, GER, 00: 26: 47-7). Closely related to the understanding of norm deviance as a problem, the interviews reveal a shared understanding of the perceived societal relevance of having an abled body. In this context, for instance Clara emphasizes how important it is to her, as a CWVI, “that people show consideration, that you know you belong, that you are accepted by the group and so on” (Clara, AT, 00:18:06-3). Marge, in turn, feels “like people get a certain view” of her “as incapable” just due to her impairment and that prejudice is something that she does not want them to have (Marge, US, 0:20:16-4). Another nuance is found in Diana’s statement that she “wouldn’t wanna trade” the general school system she is attempting “for anything” because she would “rather be in an unaccommodating school that accommodates [..her] challenges rather than sitting bored out of [..her] mind in a school geared towards kids with special needs or visual impairments“ (Diana, Pos. 126, 00:24:50-8). So, she definitely prefers the challenge of being the only CWVI in a group of ‘normal’ kids instead of sitting in a special school for the ‘non-normal’, which she apparently associates with boredom and a status of being labeled as ‘non-normal’ in a way that is visible to all. A strong discomfort of being marked as ‘not normal’ due to disability is also articulated by Robert when he says: “I hate when people go easy on me, it makes me feel lesser” (Robert, US, 0:18:09-0).

#### 3.1.2. Interpersonal and Emotional Aspects—Social Interactions

When asked to evaluate if they are feeling valued by their peers the respondents’ answers differ a lot across the sample. Some, like Lara, report that they really feel valued in PE where they “have a lot of fun” and “look out for each other” (Lara, AT, 00:02:15-1). Others like Alexander instead have a more ambivalent impression since some peers might not be “quite as enthusiastic about” learning with blind kids because “the blind people always have to be led here and there” (Alexander, GER, 00: 06: 48-3); but at the same time, he states that most of his peers “are not like that either, so they’re all very open, erm, teamwork also works quite well” (ibid.). In a more negative way, Marge has the impression that her peers see her “as incapable”, whereas she has the impression that “the system is actually the thing that’s incapable” (Marge, US, 00:20:16-4) and thus she feels confronted with a problematic education system in which she also does not feel valued by her peers at all. Another aspect highlighted by Florian is that due to his special educational needs in PE he is “very close” to his teachers and “rather not” to his peers (Florian, AT, 00:04:52-5). Similarly, Michelle states when being asked if she feels valued by her peers: “Honestly, I’m separated from my classmates a lot, so I’m honestly not that sure” (Michelle, US, 00:19:37-1). The feeling of being valued by the teachers is also assessed differently by the respondents. This ranges from the statement that “all the PE-teachers I have known here […] have been very nice […] so many of them have dealt with visual impairments” (Phillip, GER, 00: 09: 49-3), which led to an improvement in Phillip’s performance (Phillip, GER, 00: 09: 59-7), to a perhaps somewhat skeptical “I think they care, I think they want the best for me“ (Michelle, US, 0: 19:27-5), to obvious negative feelings of exclusion and infantilization, “Say, the class is playing basketball, they would kind of put me in a corner with my own ball. That way, I didn’t get hit or hurt, or anything. That was their reason for it, but personally, I think I can hear ‘Don’t play basketball’, just fine” (Lily, US, 0:02:43-4) and later, “They babied me, most of them babied me” (Lily, US, 0:23:08-5). In connection with these aspects, a certain dependence on teachers can be observed, which is not always welcome, but often seen as unavoidable. In this context, Simon reports positively how his situation in PE improved when his teacher started asking him specific questions about his vision and motor skills in class in order to address these aspects pedagogically (Simon, AT, 00:06:35-3). In contrast, Marge problematizes that during COVID-19 her “teacher used a lot of online applications or online exercise apps like Adidas Runtastic or Platform and things like that, and those really weren’t accessible” (Marge, US, 00:12:29-4). On the one hand, teachers who are attentive to the needs of these students seem to obviously provide them with a great deal of support, but on the other hand, not addressing their needs can lead to insurmountable barriers. This applies to issues of physical activity learning and lesson design as well as to social interaction in the classroom like preventing peers from making fun of disabled children (Timo, GER, 00:24:08-7). As the outlined aspects already implicitly indicate, almost all interviewees report a high relevance of their impairment for social interaction in PE. This ranges from having “a tougher time with groups, ‘cause not really many people want to play with the blind kid” (Robert, US, 0:25:42-0), to the “emotional expectations of being able to put up with a lot of bullshit” (Diana, US, 00:26:17-2), more than the “other” kids (ibid.), in combination with a perceived group separation in the form of “just me and maybe 20 to 30 other kids“ (Marge, US, 00:09:39-2), to the positive remark that dealing with the impairment in PE gives “kind of a better relationship“ because teachers “know me and who I am and stuff“ (Julian, US, 00:25:45-2).

#### 3.1.3. Didactic Aspects—Teaching Methodology and Organization of PE

In terms of methodological-didactic aspects, some students are quite positive about the adaptations they are given due to their impairment: “We have to check how this can be done and then we definitely do it like this” (Ibrahim, GER, 00:02:52-4). However, critical perspectives on adaptations and how they are (not) given seem to predominate in the sample. In this regard, some adaptations are perceived as not very conducive and how they could be better for CVWI is described: “If they could make something that makes noise in the air, that would be great” (Robert, US, 0:25:02-8). Some students report being told to do other things (Miriam, AT, 00:12:11-7), or even to go to the library instead of doing what the other kids do in PE when problems arise due to the impairment (Michelle, US, 0:16:44-3). Or they talk about being generally excluded from games (Florian, AT, 00:19:09-8). Lex even notes in general that teachers seem to have little interest in adaptations, “I feel like they don’t really go out their way to help visually impaired people, and now this could just be the schools I went to, or the teachers I had or whatever, but I feel like they don’t really go out of their way to explain... They’ll show sighted people how to do it visually, but they won’t really explain it to visually impaired people hands on, like they have... Pretty much have to” (Lex, US, 0:05:17-8). The development of adaptations is rarely described in detail by the interviewees, but in the brief comments on this, the impression is that it is mainly the teachers alone who decide whether adaptations are given or not and which are given in this case. In somewhat of a departure from this, Esther reports that her teacher decides what adaptations are given, but the students have the opportunity to voice their opinions on the decision and can sometimes decide what to do and how (Esther, AT, 00:05:56-4). However, there are definitely some exceptions, such as Ibrahim, who points out that when adjustments are made in his class, “really everyone has their say” (Ibrahim, GER, 00:11:07-6). If there are adaptations, the subjective evaluation of their quality differs considerably. Some students seem to be downright enthusiastic: “When we have PE classes, I always do my best to be there and [...] generally everything is […] adapted to the visually impaired and blind. And [...] that’s why I have a lot of fun with it and always try, and always try to do my best in class” (Philipp, GER, 00: 08: 31-8). Others, however, state that it can be difficult to find an activity that is not too boring for others and doable for them (Clara, AT, 00:08:14-6), which illustrates that adaptations are perceived as problematic and that adaptation here is seen more in terms of choosing other activities than in changing activities in their execution. Furthermore, adaptation can apparently mean going “with the adaptive kids” (Lex, US, 00:11:10-3), which seems to be implicitly associated with social relegation. In her case, Lex feels that teachers sometimes give her another activity to make her feel “like [..she] was participating, but [..she] really wasn’t” (Lex, US, 00:17:50.5). And Marge even states that she “never liked PE classes just because they weren’t really designed for [..her] participation“ (Marge, US, 00:08:08-7). The students’ statements reveal that it is not uncommon for most of them to be excluded from certain activities and sometimes even from lessons altogether. Some interviewees describe this as something positive since they are “allowed to quit the game” (Philipp, GER, 00: 03: 54-9), others emphasize that they are not really happy that activities (like basketball) that do not work for them are simply followed anyway (Esther, AT, 00:07:51-5). But others instead state that in their opinion there is no exclusion since always everyone is considered, for instance in trying “to make the games in a way that everyone can play” (Marlene, GER, 00: 02: 16-4).

### 3.2. Results of the Type-Forming Analysis

Based on the evaluative categories, each respondent was characterized by a brief description of themselves. By contrasting these brief descriptions, six clusters were identified in which the respondents each had similar characterizations. These were used in terms of a type-forming analysis [16] and six types of students were distinguished. Mainly these types differ in the way in which the main categories are described (as problematic, as positive, neutral as a given fact, or are rarely problematized) by the respondents (see Table 3). Each student was assigned to one of these six types by the research team (see Table 4), noting that some of them can be described as typical of the type to which they are assigned, while others are somewhat peripheral. For a better understanding of the types, each type is characterized below by a brief exemplary description of an interviewee typical for it.

Type 1: “No Problems”

Lara attends a vocational school specialized for CWVI in a major city in Austria. She reports that more or less everything seems to be fine for her. Apart from minor annoyances, no major problems are reported, but rather a narrative is transported that everything is OK. The extent to which problems are nevertheless hidden beneath this perhaps deliberately staged surface cannot be revealed by this analysis. Anyway, in terms of societal aspects, being normal seems to be important to Lara and she seems to share the idea that fit and able bodies are perceived as normal. This becomes visible as she describes “good moments” being those in which she was relatively able to perform some sports (Lara, AT, 00:23:25-0). In this context, the main goal of PE is seen as being fit, which is also subjectively associated with feeling good (ibid., 00:13:13-4). At her previous school (a general school), she had bad experiences of being treated as not normal, but at the current school, she feels comfortable and normal because “everyone has their own background here” (ibid., 00:06:54-7). So not being normal is apparently kind of normal here. Along with that, Lara feels valued in her actual class: “basically we are always there for each other and have fun together” (ibid., 00:03:34-4). This also includes the teacher (Lara, AT, 00:04:38-2). That is, she feels good in class, what in her perception in a positive way is depending on the way the teacher designs the lessons (ibid., 00:10:21-5), dealing in a positive way with her impairment (ibid., 00:04:38-2). Lara doesn’t talk much about didactic aspects. However, she says, that some adaptions are given to her in ball games for instance (ibid., 00:25:39-3) which she associates with fun (ibid.). These adaptions seem mainly to depend on the teachers’ initiative. But individualized adaptions seem to be rare, as basically, “everyone does the same thing most of the time” (ibid., 00:17:58-3).

Type 2: “Problematic Society”

Ibrahim attends a special school for CWVI in a medium-sized city in Germany. In the interview, he reports good adaptive teaching and very good relationships with his teachers and most of his classmates. Problematic for him seems to be a hierarchical order in which those with greater impairments are worth less due to socially transported ableist notions of normality, which leads to superstition among some of his peers. Seemingly as a matter of course, better-sighted people appear to be more normal and efficient than blind people, which is apparently implicitly justified by social notions of normality and the associated ideas of efficiency (Ibrahim, GER, 00:18:30-6). To these few students, Ibrahim attributes “arrogance” (ibid., 00:22:00-0) and he describes them as “the better sighted, who still see their thirty percent, who [...] always go like ‘oh, I’m so underchallenged’” (ibid., 00:08:37-0), which can even lead to bullying because of their ableist “mindset” (ibid., 00:19:35-7). But in general, he feels very valued in his class, both by his teachers (ibid., 00:00:33-8) and by classmates who do not have the mindset described above (ibid., 00:09:22-9). He reports that in PE, “great attention is paid to feedback from the individual participants [...] and that everyone can have their say if they have needs, suggestions for improvement, criticism, questions, etc. “(ibid., 00:02:52-4). Thus, Ibrahim has already introduced his own adaptation suggestions several times, which were also implemented, he reports proudly (ibid., 00:11:07-6). In this context, it is noteworthy for him that “it never came to the point that a game situation or a sports situation would have taken on an extent that would have excluded me there in terms of ability” (ibid., 00:11:07-6).

Type 3: “Adaptions Needed”

Michael attends a vocational school specialized for CWVI in a major city in Austria. He rarely problematizes concerns related to societal, interpersonal, or emotional aspects in the interview. But he delivers the message that for him the lessons could and should be adapted in a better way. Maybe due to COVID-19, he talks about his lessons as something that has happened in the past and never in the present. Different from most of the interviewees in the sample, Michael almost does not talk about ideas of being normal and his feelings related to these. There is only one exception, when being asked about the teachers’ expectations towards him in PE, he answers “None at all. What I’m able to do, I can do; what I’m not able to do, I cannot do” (Michael, AT, 00:11:30-2). Here, he implicitly makes clear that this lack of expectations towards him is due to his impairment. In the other parts of the interview, the impression is that he more or less avoids talking about normality or impairment. Regarding interpersonal and emotional aspects, he reports feeling completely valued by his peers, without having any negative experiences: “We felt valued by each other. Yes, all of us. And when somebody needed help, even the students helped us, this means classmates helped, we helped each other all the way” (ibid., 00:09:02-2). Interestingly, being asked about bullying he knows very well where bullying occurs in school (mainly in PE lessons and in locker rooms; ibid., 00:14:41-2), but at the same time he is keen on saying that he does not have any bad experiences in this respect (ibid., 00:10:11-8). Also, he feels accepted (and kind of dependent) on his teachers who help when problems arise and look for solutions (ibid., 00:06:43-9; 00:07:14-2). In terms of didactic and methodology aspects, Michael says that sometimes adjustments are made, but mainly he has the impression that “actually the same is done, with everyone” (ibid., 00:12:32-3). If problems arise in doing certain activities, the teachers sometimes explain them again or they let him skip these activities (ibid., 00:07:14-2). But when he reports on his balance problems in high bar gymnastics, which apparently cause him trouble, it is noticeable that he misses appropriate adaptations (ibid., 00:16:42-0).

Type 4: “Feeling OK”

Julian attends a general education school in a large city in the USA. In terms of interpersonal and emotional aspects, he reports that he is doing well, but at the same time, he sees problems both in society and in PE due to his impairment. As far as society is concerned, Julian seems to suffer on the one hand from deviating from what is considered normal (Julian, US, 0:18:20-6). This is evident, for example, when activities that he is not so good at are carried out in PE without any consideration for him (ibid., 0:06:16-3). On the other hand, he still feels somehow normal, since in his opinion “no one’s perfect” (ibid., 0:19:23-7). So, most of the time he does not report to feel too bad about socially transported ableist ideas, but sometimes he seems to suffer from them. Related to social-emotional aspects, he strongly feels valued by his peers and has the impression to be “participating fully” (ibid., 0:18:10-0). He also has the impression, that teachers care (ibid., 0:19:43-4) and have a “meaningful relationship” with him (ibid., 0:25:31-6), maybe even a “better relationship” since they know him and know who he is (ibid., 00:25:45-2). Only sometimes he implicitly gives the feeling that, in his eyes, some teachers might do more to provide full participation (ibid., 0:24:48-0). In terms of didactics, Julian explains that most of the time he does the same as his sighted classmates (ibid., 0:13:33-4). Only rarely are adjustments made and Julian misses verbal explanations since most things are explained by demonstration (ibid., 0:27:05-9). If he is unable to do some activity, he can choose not to participate (ibid., 0:14:00-9), but that seems to be a double-edged sword since grades are given mainly based on participation (ibid., 0:14:35-3). So, he has to hope that the teachers are understanding (ibid., 0:15:00-8)—which most are—but it kind of looks like a perceived burden of dependence.

Type 5: “Feeling Relatively Normal”

Serena attends a special school for CWVI in a medium-sized city in Germany. She is aware of the social meaning of normality and the impact this has on interpersonal relationships. However, because she considers herself relatively normal—and thus able (to do sports)—she reports being highly bored in the adapted PE classes she attends. In an obvious reference to socially widespread notions of normality and performance, which she associates with the PE classes she previously attended at a regular school, she describes her PE classes in the special school as “terrible” (Serena, GER, 00:02:56-8). The lessons are “much too easy and that’s not so much fun, because [...] I like this competition and that you prove yourself to others [...] and here it’s just not like that” (ibid., 00:03:19-4). The fact that it is not fun is due to the less normal students because these “have to adapt somehow and if we […] play normal soccer then not everyone can participate” (ibid., 00:05:35-6). “Most of them lack orientation completely and sometimes you feel a bit like you’re being made fun of, I mean they can’t help it, but still, it’s like, half the time it’s about where what is and how you can orient yourself and stuff” (ibid., 00:10:01-7). Regarding the teachers she notices “that they actually want us to have fun—yes—but I don’t know, anyway it’s not like that” (ibid., 00:06:56-3). As a result, she feels “a bit neglected” by her teachers (ibid., 00:10:46-2). Regarding didactic aspects, she reports some lessons in which the blind and visually impaired were separated into two groups and she was able to play “normal” field hockey and basketball, which she describes as “great fun” (ibid., 00:15:18-0). But fundamentally, she lacks performance requirements in PE, and her remarks again make clear that she strongly ties performance to ableist notions: “We don’t have to be able to do anything [...] even the blind who have a lot of problems get an A” (ibid., 00:19:25-1).

Type 6: “Concerned”

Michelle goes to a general education school in a large city in the USA. She is obviously concerned on all three levels (societal, interpersonal-emotional, and didactic). With regard to societal aspects, Michelle does not feel normal, as she distinguishes herself from her “normal peers” (Michelle, US, 0:08:35-4) and this seems to be reinforced in PE even more than anywhere else. Here, she is not really wanted to participate in many activities. It is not that she is really told by her teachers not to be able, “but I think they’ve just kind of assumed so, and they were just like, ‘You know, just go do this instead’” (ibid., 0:16:27-8). At the end of the day, teachers do not seem to make an effort. She wants to feel “like a normal” school kid (ibid., 0:10:02-0), but it looks as if she cannot. Interestingly, being asked directly, she reports feeling valued by her teachers (“I think they care”; ibid., 0:19:37-1), but the behavior she describes from her teachers and the emotional connotation suggest the opposite. She also notes that it is difficult for her to understand the feeling of being valued by her classmates because she is unpermitted to be in the same space (ibid., 0:19:37-1). If she can participate and even make her own decisions in PE seems to be strongly depending on the teacher she has (ibid., 0:18:02-5). Again, the strong influence of the fact of being impaired on interpersonal relations is visible. Related to didactics, Michelle describes adaptations as something that sometimes happens to her and rather as something she works on with her teacher. She reports to be often excluded from activities that her teachers (or aide) considered too dangerous for her, and she would rather go to the library than participate in PE (ibid., 0:32:04-9). She comments that this is “kind of frustrating because I couldn’t participate with my peers and they could do what I couldn’t” (ibid., 0:12:46-6). In telling her story she gives the impression that adaptations are not valued and that the structure of the typical game seems to be more important to teachers and other adult stakeholders than her participation.

## 4. Discussion

In this study, we examined the subjective experiences of visually impaired children, representing three different countries (Austria, Germany, and the US) and educational contexts in order to gain an understanding of barriers and challenges that influence their experiences in PE. To our knowledge, this is the first study to explore the perspectives of CWVI across three different cultural contexts and education settings, which offers interesting and important lessons about the experiences of CWVI that emerged in each international context. In exploring these experiences, several salient features that informed feelings of acceptance, belonging, and value were identified. Importantly, all three levels (societal, interpersonal, and emotional, as well as didactical) were relevant in all interviews. Even if the typology carried out revealed differences in the way of dealing with it, above all else, ableist ideals of normality appeared to be omnipresent in reflections about our participants’ experiences and were connected to relationships with peers and teachers across countries and contexts. These instances were characterized by our participants’ reflections about the importance of ‘feeling normal’ and the connection this feeling has to being good or sporty within PE spaces, as well as connections ‘feeling normal’ had to relationships with peers and teachers. Among our participants, few expressed ‘feeling normal’ within their PE classes (mainly those assigned to the types “no problems” and “feeling relatively normal”), and this was largely exclusive to CWVI were educated in segregated spaces, where CWVI were educated with only CWVI. Perhaps this is unsurprising, as these types of settings tend to be characterized as accommodating and ‘inclusive’, given that minimal or no special or individual accommodations are needed [18] since the space and place of education is already constructed with CWVI strengths and needs in mind. That is, participants may feel ‘normal’ within these spaces, as normality within those settings is constructed through students’ ability to ‘do as others are doing’ without any added support [19]. Hence, by imposing reasonable requirements and providing relative autonomy in PE some students are put into a situation where they can feel relatively able and thus normal. Being mindful, though, this was only depicted by some participants, even among those in contexts where only CWVI were educated, which speaks to the problematic nature of socially conveyed normality requirements infiltrating each cultural context and educational space, which may be more so a function of expectations around PE, sport and ‘sporty bodies’ [10] than geographic settings or educational contexts. That is, regardless of the setting, traditions of PE, such as the utilization of normative assessments in classes, for example, may still perpetuate ideals about what is and is not a sporty or capable body within a PE context, and therefore present opportunities for visually impaired students to feel ‘not normal’. This is an important finding that highlights that one’s impairment may be more powerful in influencing experiences or impacting inclusiveness than the setting or culture in which they are educated. In addition, it could be stated that those who feel normal also clearly refer to societal ideas of normality, only they experience the significance of these ideas in PE as less relevant (type “No problems”), or they see themselves as relatively normal, significantly fueled by the active comparison to those who deviate even more from the norm than they do, rather than comparing themselves with “normally abled” people (Type “Feeling relatively normal”). However, some others explicitly thematize their non-normality as problematic, but at least in class have the impression of feeling accepted and valued, which seems to make their non-normality a little easier (type “Feeling OK”).

With this in mind, and aligned with prior research exploring CWVI’s experiences within PE classes [9], most participants within this study depicted feeling ‘not normal’ during PE. According to our participants, feeling ‘not normal’ was related either specifically to their vision, or to other people’s prejudices of their (in)capabilities, and perceptions about their vision. When vision is the lynchpin for feeling ‘normal’, it supports and reinforces a medical-oriented ability-disability binary, which is perpetuated by ideals of normality rather than accepting and appreciating diversity [20]. For us, and as expressed elsewhere [10,21], combating this perception ‘normal’ = good, which was expressed by many of our participants and explicitly problematized by some (mainly type “problematic society” and “concerned”), will take a changing of perspectives from teachers, as well as teacher trainers, from a norm-based ideology to an acceptance of nonnormative-based ideologies where non-normative body conceptions and movement patterns are accepted and appreciated. It will take a collective reflection of our field to make these ideological changes in order to deconstruct ideals of ‘normal’ which trickle down from societal discourses over interpersonal relationships to how students are treated, and subsequently, how they feel about their own place within sport and PE contexts. In a connected fashion, feeling ‘normal’ was also restricted for our participants when others demonstrated a lack of understanding of visual impairment as well as prejudices on the (in)capabilities of visually impaired people (regarding society in general by type “Problematic society” and “Concerned”, in terms of PE lessons by type “Adaptions needed” and “Concerned” as well). This finding has emerged in other work focused on the experiences of visually impaired people within educational contexts [21]. For some, the answer to this barrier to subjective experiences of inclusion may be to further educate teachers on the capabilities and needs of CWVI [22]. However, this is a complex issue, given we, as a field, still know very little from an empirical standpoint on how to train teachers to work with this population of students in many, or any, contexts. In this respect, the typology in this analysis might offer some structure to better understand the multi-layered impact of social expectations of normality in PE and the many ways CWVI deals with them.

Another salient feature of our interviews was the concept of ‘feeling valued’. Feeling valued, a feature of inclusion [9] that is often underdefined in educational literature, can be thought of as a positive affective response that arises when significant others (such as teachers or peers) confirm that an individual possesses qualities that are worthy and desirable [23]. Accordingly, feeling valued has a relational element, as its emergence as a subjective experience relies on some form of praise or recognition from others about one's abilities within a given context [24]. For several of our participants, feeling (un)valued was inextricably linked to their belief that others, namely their peers, viewed them as being (in)capable within the sporty context of PE because of their impairment. This issue was exacerbated for some by their overreliance on their PE teacher throughout their class time, which further demonstrated to their peers that they were incapable or unable to participate, and therefore their skills or abilities were not valued, without the help or assistance of their physical educator. This finding has some problematic implications, as it highlights how some teaching practices that are often discussed while teaching visually impaired youth, such as having adults maintain close proximity to students, may have unseen or unspoken negative outcomes in reducing the value others see in the students’ ability. This finding may help to stimulate thinking about how and when various types of teaching practices should or could be implemented within PE contexts, regardless of the structure of that educational setting. While feeling valued vis-à-vis recognition from peers was one element among our participants, others also reflected on feeling valued by their teachers. Importantly, this paper is among the first to explore feeling valued among CWVI within PE and, as such, findings associated with PE teachers in this respect are novel and contribute to existing work. Two different avenues were presented here, where some participants described instances of feeling valued and being capable when their teacher provided adequate adaptions that allowed them to participate (mainly types “Feeling OK” and “Problematic society”); however, others noted that they were not afforded this opportunity, and needed adaptions were not provided (mainly types “Adaptions needed” and “Concerned”). In these instances, feeling valued was not afforded to these students as they believed the teacher did not care to think about and construct adaptions that could make teaching and learning meaningful within this context for them. Again, to reiterate, since feeling valued is a relative concept that is dependent on others [24], the teachers’ behaviors in this instance did more to make visually impaired students feel dependent and paternalized than valued and capable. Few others (type “Relatively normal”), however, felt primarily underchallenged because of the adjustments and rather urged instruction that was more in line with their better abilities. Here, it becomes clear that in these contexts an individually appropriate balance of adaptation and demand is required. Achieving this balance is a highly complex pedagogical task that can only be solved on a situational basis and requires professional staff who not only have the necessary knowledge but also the pedagogical skills to do so.

Interestingly, while the setting in which the participants were educated in may have had some impact, we would assert that this impact was not overly strong and that each of our participants’ narratives helped to demonstrate how they could not escape the power of social distinctions and related and existing hierarchies. That is, in most instances, the geographic context in which the participants were educated, as well as the setting within this context, did not appear to be a primary indicator for the participants’ experiences, suggesting that societal perceptions and hierarchies related to visual impairment, which have been identified and described in PE research previously [9], may have stronger influences on subjective and lived experiences in PE. This, again, is an important finding that is unique to this ‘first of its kind’ study that explored feelings in PE across context and culture. One exception for this, though, may be related to participants from the US context. That is, while participants from Austria and Germany were well distributed across several typologies of experiences, as noted in Table 2, most (5 of 7) participants in the US context were categorized in the ‘concerned’ category. This finding may speak to the numerous challenges and barriers that have been, and continue to be, expressed among visually impaired mainly female students in the US about the marginalization, ostracization, and discrimination they experience within integrated PE contexts in that country [8]. Of note though, we must be cautious with attributing these concerns to either the geographic location or school context specifically, given that we do not have data in this particular study describing research in other settings (self-contained, schools for the blind) in the US or integrated classes within Austria or Germany. For us, these findings support the need for further research more deeply exploring the needs of visually impaired students within each of these geographic locations and contexts to support their capabilities and needs, while reducing apparent and voiced challenges and barriers.

## 5. Conclusions

This study provides insight into the barriers and challenges that influence the PE experiences of CWVI representing three different countries and distinct educational contexts. Importantly, we learned that regardless of the geographic context or educational space in which our participants were educated, ideals of normality and associated hierarchies provided a framework for their experiences. That is, all participants reported that they were forced to exist within PE spaces and places where social hierarchies existed and that normality requirements or expectations within spaces dictated their abilities or capabilities to be or appear successful. Some seemed to gain feelings of being valued by seeing themselves as relatively normal either compared to their peers with more severe impairments or within their “safe space” in PE where everyone has an impairment. But the idea of normality and associated values remain for all of them as the sword of Damokles. Hence, some might find feelings of being valued in a more adapted PE (which is a lot!), but still, they know that socially they’re somehow less valued due to their impairment. They “just” seem to find different ways to deal with this issue or draw diverging lines of differentiation, as the various types in the analysis illustrate. An inclusive pedagogy should and must address these societal aspects as well—this can be named as a major task for the future.

While findings like these have emerged in other research focused on one specific geographic context or educational space [9], this is the first study, to our knowledge, that has demonstrated the existence of these features of education across geographic and educational spaces and places. While these features of PE existed across contexts, our participants appeared to deal with them differently. For example, some participants noted that they did not feel normal and were, rather, below or on the bottom of any hierarchy that existed within the class. As highlighted in other work, these students did not believe there was anything that was or could be achieved because they simply could not participate due to their impairment or because of socially transported ableist norms of society [9]. For others, feeling normal was available to them, but being educated in self-contained contexts where activities were ‘too easy’ made PE experiences unenjoyable or ‘terrible’. In these instances, it appears that some of our participants were enjoying a privileged position in the social hierarchy because of their relative capabilities within activities, and that specific didactic considerations were needed (such as dividing classes into different activities) to keep their attention. As such, it appears that while social hierarchy and normality requirements were present throughout the experiences, our participants presented a proverbial salad of experiences within these contexts and spaces.

In concluding, it should be noted that there are obvious limitations to this study. As mentioned, we interviewed three groups of visually impaired students from different geographic contexts (i.e., Austria, Germany, the US), and within each of these contexts, the school environment was unique. Because of this, we cannot make direct comparisons between participants across regions or school environments to make claims about one being able to provide ‘better’ or ‘worse’ experiences, and we cannot make direct claims about any geographic context or educational setting as being more or less prepared to educate visually impaired students. What we can do, and believe that we have, is to deeply unpick the subjective experiences of visually impaired participants who engaged in PE classes across settings and countries to demonstrate similarities in their experiences despite differences in context. We would be remiss, as well, not to mention that our participants were somewhat homogenous in terms of impairment, where none of our participants identified as experiencing any additional impairment beyond visual impairments. As such, the voices of those who experience visual impairments with other comorbid impairments are silenced within these findings, and we encourage future work where those with additional disabilities are included, such as deafblind children, are included within research like this to amplify their voices about their PE experiences.

## Figures and Tables

**Figure 1 ijerph-20-07081-f001:**
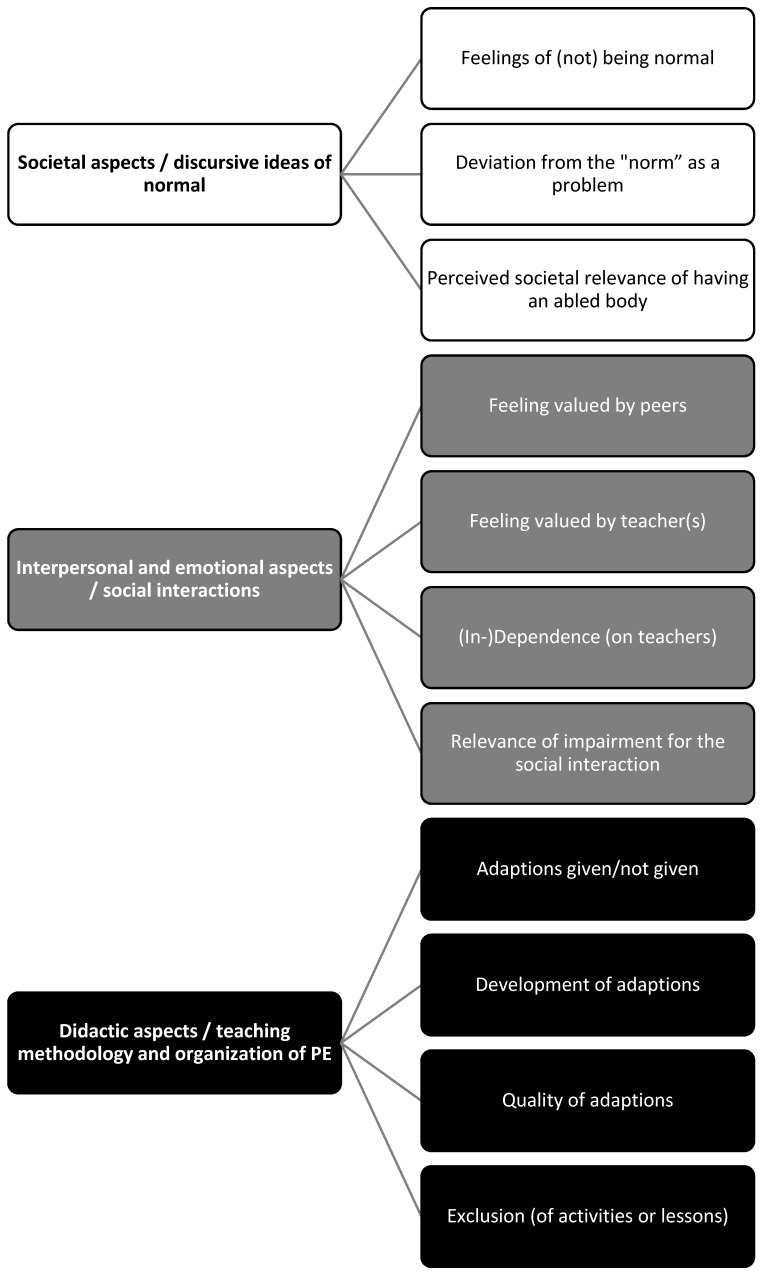
The category system of the analysis.

**Table 1 ijerph-20-07081-t001:** Sample.

Interviewee	Country	Gender	School Type	Interview Setting	Age/Grade *
Clara	Austria	female	specialized vocational school	single interview by telephone	21 years
Esther	Austria	female	specialized vocational school	single interview by telephone	16 years
Florian	Austria	male	specialized vocational school	single interview by telephone	19 years
Lara	Austria	female	specialized vocational school	single interview by telephone	no information
Michael	Austria	male	specialized vocational school	single interview by telephone	17 years
Miriam	Austria	female	specialized vocational school	single interview by telephone	no information
Simon	Austria	male	specialized vocational school	single interview by telephone	16 years
Alexander	Germany	male	special needs high school	tandem interview with Marlene, face to face	10th grade
Giorgio	Germany	male	special needs high school	tandem interview with Timo, face-to-face	10th grade
Ibrahim	Germany	male	special needs high school	single interview, face-to-face	10th grade
Marlene	Germany	female	special needs high school	tandem interview with Alexander, face to face	10th grade
Pablo	Germany	male	special needs high school	tandem interview with Serena, face to face	10th grade
Phillip	Germany	male	special needs high school	single interview, face-to-face	10th grade
Serena	Germany	female	special needs high school	tandem interview with Pablo, face to face	10th grade
Timo	Germany	male	special needs high school	tandem interview with Giorgio, face to face	10th grade
Diana	US	female	general high school	single interview by video conference	15 years old
Julian	US	male	general high school	single interview by video conference	14 years old
Lex	US	female	general high school	single interview by video conference	15 years old
Marge	US	female	general high school	single interview by video conference	14 years old
Michelle	US	female	general high school	single interview by video conference	13 years old
Robert	US	male	general high school	single interview by video conference	15 years old
Thomas	US	male	general high school	single interview by video conference	12 years old

* To protect anonymity, age was not specified in this setting and only school grade was given. For the tenth graders surveyed in this setting, the age is likely to be between 15 and 17 years.

**Table 2 ijerph-20-07081-t002:** Categories with anchor examples.

Category	Subcategory	Anchor Example
Societal aspects/discursive ideas of normal	Feelings of (not) being normal	“At that moment you just wish you were a normal child.” (Clara, AT, 00:11:18-9)
	Deviation from the “norm” as a problem	“We actually came to this school in order to no longer have this disadvantage with this vision” (Giorgio; GER, 00:11:09-4)
	Perceived societal relevance of having an abled body	“I feel like people get a certain view of me that I don’t want them to have. They sort of see me as incapable” (Marge, US, 0:20:16-4)
Interpersonal and emotional aspects/social interactions	(Not) Feeling valued by peers	“In PE, because we’re not that many people, we actually always have a lot of fun. We also look out for each other, so if someone really can’t do it, then we help him or her; or if I don’t see something, for example, or if it doesn’t work for me, then I say, can we change it like this? And that actually works quite well” (Lara, AT, 00:02:15-1)
	(Not) Feeling valued by teacher(s)	“I think they care, I think they want the best for me” (Michelle, US, 0:19:27-5)
	(In-)Dependence (on teachers)	“And then it happens that some people make fun of them. But the teachers actually try to prevent that directly” (Timo, GER, 00:24:08-7)
	Relevance of impairment for the social interaction	“…sometimes I have a tougher time with groups, ‘cause not really many people want to play with the blind kid” (Robert, US, 0:25:42-0)
Didactic aspects/teaching methodology and organization of PE	Adaptions given/not given	“I feel like they [teachers] don’t really go out their way to help visually impaired people, […] They’ll show sighted people how to do it visually, but they won’t really explain it to visually impaired people hands on, like they have—pretty much have to” (Lily, US, 00:05:17-8)
	Development of adaption	“Anyway, they are people who are very open to ideas and suggestions from us students, which I really like. And what I also like about our PE teachers is that they care about each and every one of us in PE class” (Alexander, GER, 00:04:31-2)
	Quality of adaptions out of the individual perspective	“When we have physical education, then I always do my best to be there […] generally, everything is adapted to the blind, thus adapted to the visually impaired and blind. And that’s why […] I have a lot of fun with it and always try, and always try to do my very best in class” (Phillip, GER, 00:08:31-8)
	Exclusion (of activities or lessons)	“It was like this from the beginning, that they said, no I am not allowed to participate” (Florian, AT, 00:19:09-8)

**Table 3 ijerph-20-07081-t003:** Characteristic of the types.

Type	Societal Aspects	Interpersonal and Emotional Aspects	Didactical Aspects
“No problems“	Neutral as a given fact	Described as positive	Rarely problematized
“Problematic society“	Described as problematic	Described as positive	Described as positive
“Adaptions needed“	Rarely problematized	Described as positive	Described as problematic
“Feeling OK“	Described as problematic	Described as positive	Described as problematic
“Relatively normal“	Neutral as a given fact	Described as problematic	Described as problematic
“Concerned“	Described as problematic	Described as problematic	Described as problematic

**Table 4 ijerph-20-07081-t004:** Distribution of types in the sample.

Type	Respondents Assigned	Female	Specialized School	AT	GER	US
“No problems“	3	2	3	2	1	-
“Problematic society“	3	-	3	-	3	-
“Adaptions needed“	3	-	2	2	-	1
“Feeling OK“	4	1	3	1	2	1
“Relatively normal“	2	1	2	-	2	-
“Concerned“	7	6	2	2	-	5
Total	22	10	15	7	8	7

## Data Availability

Data is unavailable due to privacy or ethical restrictions.

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
