# Peer review of "Barriers and Challenges for Visually Impaired Students in PE—An Interview Study with Students in Austria, Germany, and the USA"

_ijerph, 2023, doi:10.3390/ijerph20227081_

Round 1
Reviewer 1 Report
Comments and Suggestions for Authors
The manuscript contains some interviews with visually impaired students from three different countries.
The number of people interviewed is small, 22 over all the world, and the conclusions are consequently meaningless.
From my point of view the manuscript is devoid of scientific significance. According to a report by the World Health Organization, there are currently 284 million people in the world who are visually impaired, and 39 million people are blind. It is well evident that 22 interviews of students with visual impairments do not constitute a sufficient data base to elaborate useful statistics and develop guidelines to design aids for blind or visually impaired subjects.
The content is not publishable in a scientific journal.
Author Response
Dear reviewer,
thank you for taking the time to review our article. Regarding your comments we With regard to your comments, we would like to explain that we deliberately conducted a qualitative and thus not a representative survey. Hence, the goal of this qualitative study was not to gain insights into the situation of CWVI using a representative sample. Rather, we want to understand the ways in which CWVI perceive barriers and challenges in PE and how they position themselves in relation to them. In this regard, the sample, which consists of CWVI in different countries and school settings, allows us to identify basic structures and different ways of dealing with them. Thus, the aim of the study was not to capture the situation of all CWVI in the world – in the sense of a social reality to be uncovered – but rather to understand basic interrelationships and dynamics of social structures (conceived as changeable structures) and thus develop theory. We hope that this approach is now even better elaborated in the revised version.
Reviewer 2 Report
Comments and Suggestions for Authors
Overall, this is a well performed study which provides insights into the barriers and challenges that visual impaired students experience in physical education classes. What makes this study standing out is the fact that the authors have collected data from 3 different countries each with different approaches to teaching and dealing with students with visual impairments.
I only have some minor suggestions to the authors:
Line 15: after the comma the sentence that follows ", not least of terms of healthy and self-determined development of our time” it’s not fluent and the meaning that the authors want to transfer to the readers is not understandable. Please revise the whole sentence.
Line 21: you are providing an abbreviation “CWVI” without first defining it
Table 1: typing mistake its “School type” not “Schol type”.
Author Response
We thank the reviewer for his appreciation of the article and thorough review of the manuscript. We have edited all of the monita. They are listed below.
“Line 15: after the comma the sentence that follows ", not least of terms of healthy and self-determined development of our time” it’s not fluent and the meaning that the authors want to transfer to the readers is not understandable. Please revise the whole sentence.”
Thank you for this comment. We revised the sentence.
"Line 21: you are providing an abbreviation “CWVI” without first defining it”
Thank you for this important note. We have revised this throughout the text for stringency.
“Table 1: typing mistake its “School type” not “Schol type”.”
Thank you, we have corrected that.
Reviewer 3 Report
Comments and Suggestions for Authors
This manuscript explored the barriers and challenges for visually impaired students by an interview study. It is well-structured. I think that the article can be published after making some modifications:
1 - It is recommended to provide a complete spelling “Children with Visual Impairments"before using the abbreviation "CWVI".
2 - Figure 1 is important for this paper, while the iterms in it lack explain or citation. How to get it?
3 - The section of 3.1 looks a little bit redundant about the feedback from participants. Their viewpoints can be further refined.
4 - The six types of persona are one of contribution of this study. It is not appropriate to directly use a participent because individual characteristics overlap, while persona reveal a common characteristic of group. How about directly listing features?
5 - In conclusion, it might be useful to explain more in detail how your research has contributed to the theory and practice of the issue under consideration.
Comments on the Quality of English LanguageThe quality of English language is fine.
Author Response
We thank the reviewer for his appreciation of the article and thorough review of the manuscript. We have edited all of the monita. They are listed below.
“1 - It is recommended to provide a complete spelling “Children with Visual Impairments"before using the abbreviation "CWVI".”
Thank you for this important note. We have revised this.
“2 - Figure 1 is important for this paper, while the items in it lack explain or citation. How to get it?”
The points in Figure 1 are the categories of the analysis. These arise both from prior knowledge (as described in Chapter 1) and inductively from the material (as described in the methods section). The main categories can be seen as a main division into social aspects on the macro and micro level as well as on the instructional level. For a better understanding we have included some information on this in the methods and the results section.
“3 - The section of 3.1 looks a little bit redundant about the feedback from participants. Their viewpoints can be further refined.”
We have reworked this part a little bit to provide a better understanding.
“4 - The six types of persona are one of contribution of this study. It is not appropriate to directly use a participent because individual characteristics overlap, while persona reveal a common characteristic of group. How about directly listing features?”
Thanks for bringing up this relevant point. To avoid confusion of individual characteristics with characteristic features of the 6 types, we have added information about the type-analysis as well as Table 3 for outlining the main characteristics of the typology. However, we believe that the exemplary portraits of the respondents bring the established typology to life and provide a deeper understanding of the interrelationships of the categories under study. With the information now available, it is possible to distinguish between individual features and characteristic features for the respective types.
“5 - In conclusion, it might be useful to explain more in detail how your research has contributed to the theory and practice of the issue under consideration.”
Thanks for this comment. We have added some clarifications in the conclusion to make this clearer.
Round 2
Reviewer 1 Report
Comments and Suggestions for Authors
the manuscript was not significantly improved respect to the previous version.
the content is still not sufficient for the publication for the same reasons explained in the revision of the previous version.
Author Response
Thank you for looking at our paper again. Since your main point of criticism is aimed at the non-existent representativeness of the sample, we would like to emphasize that we are pursuing a qualitative research approach in which it is expressly not the aim to uncover a reality per se via a representative sample. Our aim is to understand the situational contexts in depth and, on this basis, to identify any structural similarities. This research is therefore aimed at generating theory and less at verifying or falsifying theoretical assumptions. To clarify this, we have inserted the following passage in the text:
"The observation of these situational manifestations of social reality in the deliberately contrastive settings was therefore not intended to reveal reality per se via a representative sample. Rather, the aim of the study was to identify structural similarities between the situational manifestations and to understand individual characteristics with regard to these structural similarities."
Reviewer 3 Report
Comments and Suggestions for Authors
I agree to publish this manuscript.
Author Response
Many thanks for the positive feedback and constructive comments in the first round of reviews.